# A Systematic Review Exploring Dietary Behaviors, Psychological Determinants and Lifestyle Factors Associated with Weight Regain After Bariatric Surgery

**DOI:** 10.3390/healthcare12222243

**Published:** 2024-11-11

**Authors:** Nora A. Althumiri, Nasser F. Bindhim, Saja A. Al-Rayes, Arwa Alumran

**Affiliations:** 1Informed Decision Making (IDM), Riyadh 13303, Saudi Arabia; nd@idm.sa; 2Sharik Association for Research and Studies, Riyadh 13302, Saudi Arabia; 3Health Information Management and Technology Department, College of Public Health, Imam Abdulrahman bin Faisal University, Dammam 31441, Saudi Arabia; salrayes@iau.edu.sa (S.A.A.-R.); aalumran@iau.edu.sa (A.A.)

**Keywords:** behavioral changes, psychological factors, weight regain, bariatric surgery, systematic review

## Abstract

Background: Weight regains after bariatric surgery presents a significant challenge to long-term weight management and overall health outcomes. While clinical and surgical factors influencing bariatric outcomes have been extensively reviewed, there remains a notable gap in understanding how behavioral factors—specifically dietary habits, psychological influences, and lifestyle choices—impact long-term weight maintenance. This systematic review addresses this gap, highlighting the unique role of these behavioral factors in sustaining weight loss post-bariatric surgery. Aim: This systematic review aims to explore the behavioral factors—including dietary, psychological, and lifestyle influences—associated with weight regain after bariatric surgery. Methods: A comprehensive search was conducted across multiple databases, focusing on peer-reviewed articles published in English. We included observational studies exclusively focused on adults who experienced weight regain after bariatric surgery, with an emphasis on behavioral factors. Searches were conducted in MEDLINE via PubMed, Embase, Web of Science, Scopus, and Google Scholar, with the last search completed on 10 July 2024. Studies published within the past 10 years were considered. The Joanna Briggs Institute (JBI) critical appraisal tool for cross-sectional studies was used to assess the risk of bias. A total of 16 studies met the inclusion criteria: 12 cross-sectional studies, 3 cohort studies, and 1 in-depth interview study. Results: The review found inconsistencies in the definitions of weight regain across studies. Nonetheless, three primary behavioral factors were identified as potentially contributing to weight regain: dietary non-adherence and eating patterns, psychological determinants, and lifestyle behaviors. Dietary non-adherence included high-calorie food consumption, excessive intake of sweets, carbohydrates, and sweetened beverages, and low protein intake. Psychological factors such as depression, anxiety, emotional eating, and binge eating were prevalent among individuals experiencing weight regain. Lifestyle behaviors, including physical inactivity, smoking, and sedentary habits, also played significant roles. Conclusions: Post-bariatric weight management is multifaceted, necessitating a comprehensive approach that addresses dietary, psychological, and lifestyle factors. Most studies on behavioral factors linked to weight regain were cross-sectional with small sample sizes, limiting the ability to infer causality. Future research should include detailed information on dietary adherence, standardized tools to assess physical activity and sedentary behavior, and validated measures for psychological health. Tailored interventions and continuous support from healthcare professionals are essential for maximizing the effectiveness of bariatric surgery, promoting sustainable weight loss, and enhancing overall health outcomes.

## 1. Introduction

Bariatric surgery, recognized as the leading approach for weight reduction, encompasses several procedures, including gastric bypass, sleeve gastrectomy, biliopancreatic diversion with duodenal switch, and single-anastomosis duodenale bypass with sleeve gastrectomy [1]. Among these procedures, gastric bypass and sleeve gastrectomy are the most frequently performed surgeries [1]. The surgical approach is typically considered when traditional strategies, such as diet and exercise, fail to yield results or when a person’s health is severely at risk due to obesity [2]. Some of these surgeries work by limiting the amount of food one can consume, while others reduce the body’s ability to absorb nutrients [3,4]. There are also procedures that employ a dual strategy, combining both restriction and malabsorption [4].

Bariatric surgery is recognized as a transformative intervention for weight loss, improving or mitigating diabetes, and reducing the risk of cardiovascular issues [4]. Yet, a dominant issue is weight regain, which may occur soon after the surgery. The percentage of bariatric procedures deemed “failures” varies based on the criteria used to define failure and the length of the follow-up period. Significant weight regains following initial successful weight loss are common issues. Research indicates that a substantial portion of patients may experience some level of weight regain over the long term, with estimates ranging from 10% to 20% of the lowest weight achieved [5]. Clapp et al., in their systematic review, found the prevalence of weight regain to be 27.8% (ranging from 14% to 37%) seven years following surgery [6]. Although no specific definition exists in the literature for clinically significant weight regain, it can be described as progressive weight gain after a successful initial weight loss. Furthermore, inadequate weight loss can be defined as achieving less than 50% excess weight loss at 18 months post-bariatric surgery [7]. This standardized approach provides a clearer framework for assessing weight outcomes in post-surgical patients, aiding in the comparison and interpretation of long-term results [7]. However, the reported prevalence of weight regain varies significantly based on the definition used, making it challenging to determine its true rate due to diverse methodologies and definitions for significant weight regain across studies. This weight gain can lead to the reappearance of obesity-related health problems and a decline in the patient’s quality of life [8,9].

Weight control strategies in the general population typically involve lifestyle modifications, including diet, exercise, and behavioral changes to create a caloric deficit [9]. However, weight control in post-bariatric surgery patients is more complex, given physiological changes such as altered digestion, nutrient absorption, and hormonal shifts that affect hunger and satiety [10,11,12]. Additionally, psychological factors, such as body image concerns and ingrained behavioral habits, often intensify following surgery, requiring specialized interventions beyond those used in the general population [13,14]. Unlike typical weight management strategies, post-bariatric patients face unique dietary challenges (e.g., food intolerances and reduced meal portions) and psychological difficulties (e.g., disordered eating patterns and emotional eating). These distinct challenges highlight a gap in the literature on behavioral strategies tailored for long-term success after bariatric surgery.

Multiple studies have identified a variety of factors contributing to weight regain after bariatric surgery. These factors can be categorized as either non-modifiable or modifiable factors. Non-modifiable factors include surgical and clinical aspects, such as the technique used and any complications arising from the procedure, as well as physiological factors like metabolic adaptation, hormonal changes, and nutrient absorption. On the other hand, modifiable factors are primarily behavioral, encompassing dietary and eating habits, physical activity, and levels of sedentary behavior. Psychological factors, including mental health challenges and lack of support, also significantly impact weight regain outcomes.

Most systematic reviews and meta-analyses to date have focused on clinical and surgical factors affecting bariatric surgery results. However, there is a notable lack of reviews investigating behavioral factors, such as dietary habits, psychological influences, and lifestyle choices, associated with weight regain. This gap points to an incomplete understanding of the impact of behavior on long-term weight management after bariatric surgery. To our knowledge, only two systematic reviews have specifically examined these behavioral factors. The first, a scoping review, identified poor dietary adherence, maladaptive eating behaviors, insufficient follow-up, and low physical activity as contributors to weight regain [9]. The second review highlighted the prevalence of grazing behavior, reporting that it ranged from 16.6% to 46.6% and was significantly associated with weight regain in four out of the five studies evaluated [10]. While two previous reviews have explored aspects of weight regain following bariatric surgery, they lack a comprehensive analysis that specifically integrates behavioral, psychological, and lifestyle factors. These factors are critical for understanding the full spectrum of influences on long-term weight outcomes but are often addressed separately in the literature.

We recognize that this topic has been addressed in previous studies; however, our objective is to consolidate all behavioral factors contributing to weight regain into a comprehensive review. This systematic review seeks to address this gap by thoroughly examining dietary, psychological, and lifestyle factors associated with weight regain in post-bariatric surgery patients. Through this approach, we aim to provide insights that will enhance our understanding and inform tailored strategies for effective long-term weight management.

## 2. Material and Methods

This systematic review adhered to the guidelines and protocol specified in the Preferred Reporting Items for Systematic Reviews and Meta-Analyses (PRISMA) statement [10].

### 2.1. Search Strategy and Selection Criteria

To conduct a comprehensive literature search, we utilized five independent databases: MEDLINE via PubMed, Embase, Web of Science, Scopus, and Google Scholar. Additionally, we manually searched the reference lists of the included studies. The last search was conducted on 10 July 2024. Descriptors were identified using Medical Subject Headings (MeSH) for terms such as “Bariatric surgery”, “Behavior”, and “Weight Regain”. These terms were later combined using the Boolean operator AND, while their synonyms were linked using the Boolean operator OR. We included all types of studies focusing on behavioral and weight regain after bariatric surgery and published within the past 10 years. The full search methodology is provided in the Appendix A.

### 2.2. Eligibility Criteria

We included observational studies only, specifically focusing on cohort, cross-sectional, and in-depth interview designs that examined behavioral factors associated with weight regain after bariatric surgery. Interventional studies, including randomized controlled trials (RCTs) and clinical trials, were excluded from this review to ensure a consistent focus on naturally occurring behavioral patterns without imposed interventions. Systematic reviews, case reports, case series, and animal or in vitro experimental studies were also excluded to maintain a focus on primary observational data relevant to post-bariatric surgery outcomes in adults. The systematic review focused exclusively on articles that included adults aged 18 years or older who had undergone bariatric surgery and reported weight gain related to behavioral factors.

### 2.3. Study Selection

Two researchers independently and simultaneously conducted the search. They then prescreened the titles and abstracts against the inclusion and exclusion criteria. A list of potentially relevant citations was compiled, and the full texts of these sources were obtained. The researchers maintained a record of all excluded studies, noting the specific reasons for exclusion. These authors also independently evaluated the full-text articles against the listed inclusion criteria. Any disagreements were resolved through consultation and discussion. The author, identified by their initials ‘NAA,’ reverified the final list of included studies against the inclusion and exclusion criteria to ensure consistency and accuracy.

### 2.4. Data Extraction

Data were extracted from the full-text versions of all included studies. To create the extracted table, items included the authors’ last name, date of publication, reference number, study design, time since the surgery, and a summary of results.

### 2.5. Quality Assessment

To assess the risk of bias, we utilized the Joanna Briggs Institute (JBI) critical appraisal tool, which is primarily designed for cross-sectional studies [11]. The studies were categorized into low, moderate, or high risk of bias based on the JBI criteria, with each study’s risk level determined by the number of criteria met, following JBI guidelines. Given that our review included a range of study designs—cross-sectional, cohort, and in-depth interviews—we adapted this tool to evaluate each study type as consistently as possible. Although the JBI tool is not specifically designed for cohort or qualitative research, we applied relevant criteria to assess the quality of reporting and methodological rigor. We acknowledge the limitations of this approach, particularly for evaluating qualitative methodologies, and recommend that future research use tools tailored to each specific study design to ensure the most accurate quality assessment.

Each study’s risk level was independently assessed by two reviewers. Disagreements were resolved through discussion or, when necessary, by involving a third reviewer. Studies were categorized as having a low risk of bias if they met a total score of 9, a moderate risk if the score ranged from 6 to 8, and a high risk if the score was 5 or lower. In the context of weight regain after bariatric surgery, all included studies were found to have a moderate risk of bias, with 87.5% (87.5) scoring 7 or above and 12.5% scoring 6 (n = 2).

### 2.6. Data Analysis and Findings Synthesis

In this review, we employed a narrative synthesis approach to analyze and summarize the findings of the included studies. Given the variation in study designs, sample sizes, follow-up periods, and definitions of weight regain, it was not feasible to conduct a meta-analysis. This heterogeneity would have limited the reliability and interpretability of any combined statistical analysis. Instead, we categorized and summarized the results based on thematic factors contributing to weight regain, including dietary behaviors, psychological determinants, and lifestyle factors. We then identified common patterns and unique findings across studies to provide an integrated understanding of the behavioral factors linked to weight regain post-bariatric surgery. No statistical methods were used to combine results quantitatively due to the diversity of methodologies and measurement tools across studies. This narrative approach allowed us to comprehensively review the behavioral contributors to weight regain and highlight areas for future research that may support meta-analytic evaluation once more standardized study designs and outcomes are available.

In addition to narrative synthesis, we employed a thematic analysis approach to analyze the findings across the included studies. Thematic analysis was chosen to identify and categorize recurring behavioral, psychological, and lifestyle factors contributing to weight regain post-bariatric surgery. This method involved systematically coding the findings of each study and grouping them into overarching themes, such as dietary non-adherence, psychological determinants, and lifestyle behaviors. This approach allowed us to draw connections across studies, identify patterns, and provide a structured synthesis of factors influencing weight regain. By utilizing thematic analysis, we ensured that the review captured the complexity of behavioral factors involved in weight regain, allowing for a more comprehensive understanding of the main contributors and their interactions.

## 3. Results

### 3.1. Literature Search

The initial search yielded 1652 results. After removing duplicates, the titles and abstracts of 1304 articles were screened, with 16 potential studies selected for full-text reading. The exclusion of the 1288 articles was due to them being reviews, book chapters, or not focusing on behavior and weight regain. Figure 1 illustrates the flow diagram of the article selection process. Ultimately, 16 full-text articles were included in the final synthesis of this review. We extracted information on each study, including the year of publication, design, aim, and summary of results, and then we categorized them into themes.

### 3.2. Study Characteristics

Table 1 provides an overview of the characteristics of the studies included in this review. The combined sample size across studies was 8545 participants, with follow-up periods ranging from 6 months to 14 years post-bariatric surgery. Among the studies, three employed a longitudinal design [12,13], twelve were cross-sectional [9,12,14,15,16,17,18,19,20,21,22,23,24,25,26], and one utilized an in-depth interview approach [25]. The majority of participants had undergone either Roux-en-Y Gastric Bypass (RYGB) or Sleeve Gastrectomy (SG), with only one study including individuals who received Laparoscopic Adjustable Gastric Banding (LRYGB). Definitions of weight regain varied across studies, with weight regain described as any increase from the lowest achieved weight (nadir), an increase in the BMI category, or excessive weight regain exceeding thresholds of 10%, 15%, 20%, or 25% of the initial weight lost. Notably, one study did not specify a definition for weight regain [25]. This variability in definitions highlights the challenges in standardizing weight regain metrics across the literature.

Weight regains following bariatric surgery is influenced by a complex interplay of factors stemming from various behaviors, including physical activity, dietary intake, eating patterns, psychological factors, and other behavioral characteristics. In the following sections, we provide an in-depth critical analysis of each of these factors. Despite variations in the definition and measurement of weight regain across studies, the behavioral contributors to weight regain have been organized into thematic categories. Our conceptual framework in identifies three primary domains associated with weight regain - also illustrated (Figure 2):**Dietary Non-Adherence and Eating Patterns**: Patterns of non-adherence to recommended dietary guidelines, including grazing, binge eating, and inconsistent meal structures.**Psychological Determinants**: Factors such as emotional eating, depression, anxiety, and other mental health-related influences.**Lifestyle Behaviors**: This category includes physical inactivity, sedentary behavior, smoking, sleep habits, and self-monitoring practices like self-weighing.

#### 3.2.1. Dietary Non-Adherence and Eating Patterns

Of the 16 studies examining diet-related behaviors, nearly all highlighted the significant influence of dietary non-adherence and eating behaviors on weight regain. This theme was further divided into two primary subcategories:(1)**Food Intake**: Contributing factors to weight regain include cravings for sweet foods, excessive carbohydrate intake, increased consumption of carbonated and artificially sweetened beverages, high intake of fast food, and low protein consumption.(2)**Eating Behaviors**: Weight regain is associated with factors such as food addiction, uncontrolled eating, lack of knowledge regarding post-surgical nutritional needs, night eating, grazing, unanticipated weight management challenges, picking or nibbling, irregular meal patterns, increased portion sizes, and preference for energy-dense foods. Additionally, limited access to support and nutritional education often delayed participants’ ability to effectively prevent weight regain, leading some to adopt restrictive eating or dieting practices that were not conducive to sustained weight loss.

#### 3.2.2. Psychological Determinants

Mental health issues, including depression and anxiety, along with maladaptive eating behaviors such as binge eating and emotional eating, were identified as primary psychological determinants associated with weight regain. Four studies reported higher levels of depressive symptoms among individuals who experienced weight regain [13,15,19,21]. For example, a study reported that patients with persistent depressive symptoms had a significantly higher incidence of weight regain (≥20%), with 86.7% of these patients regaining weight compared to 52.0% among those without depressive symptoms, a statistically significant difference (*p* = 0.026). This result was corroborated by other studies, where self-reported depression and anxiety were significantly more prevalent in individuals who experienced weight regain (*p* = 0.008 and *p* = 0.001, respectively) [15,27]. Another study emphasized that severe postoperative depressive symptoms were a key factor contributing to greater weight regain, highlighting the complex relationship between psychological health and weight management following surgery [13].

Additionally, eating behaviors associated with psychological factors, such as emotional eating and binge eating, were strongly correlated with weight regain [9,17,18,22]. For instance, among patients who experienced significant weight regain, approximately 44% reported emotional eating, while 5% reported binge eating behaviors [22].

#### 3.2.3. Lifestyle Behaviors

Seven studies examined the impact of lifestyle factors on weight regain, including physical activity, smoking habits, self-weighing, sedentary lifestyle, sleep, and mindful eating. Four studies highlighted the positive role of physical activity in reducing weight regain. For instance, individuals who engaged in regular physical activity post-surgery were consistently less likely to experience weight regain than those who were less active or inactive [9,15,16,28]. In one cohort study, bariatric surgery patients were divided into active and inactive groups. Initially, both groups were classified as having severe obesity before surgery [9]. However, at the two-year follow-up, participants in the physically active group were reclassified as overweight, whereas those in the inactive group remained classified as having severe obesity [9]. Although physically active individuals demonstrated lower rates of weight regain, the frequency of regain was still notable across participants, which could potentially lead to the recurrence or exacerbation of comorbid conditions associated with excess weight [9].

The follow-up period for the studies included in this review ranged from 6 months to 14 years post-surgery, with significant variability in the timing and extent of weight regain reported. For instance, weight regain of 10–20% was commonly observed within the first two years in several studies, while others documented similar regain levels occurring gradually over five or more years. This distinction is critical, as the timeframe over which weight regain occurs may influence both the clinical implications and the need for intervention. Weight regains occurring within the first year, for example, suggest different behavioral or physiological challenges compared to gradual regain over a decade.

Additionally, several studies highlighted the impact of sedentary behaviors, such as prolonged periods of reading, watching TV, or using computers, which were strongly associated with increased weight regain [15,24]. A study by King et al. found that the least sedentary individuals spent fewer than 2 h per day on leisure activities involving computers or TV, compared to the most sedentary individuals, who spent at least 4.5 h. Univariate analysis revealed a significant negative correlation between self-reported moderate-to-vigorous intensity physical activity and weight regain. When physical activity and sedentary behavior were analyzed together, sedentary behavior alone maintained a significant association with weight regain [13].

#### 3.2.4. Integrated Influence of Dietary, Psychological, and Lifestyle Factors

The six studies collectively underscore the compounded risk of weight regain when dietary non-adherence, psychological issues, and lifestyle factors are present simultaneously. For example, individuals with depressive symptoms and high levels of sedentary behavior were particularly susceptible to weight regain, especially if they struggled with dietary control and emotional eating. One study highlighted that patients facing unexpected weight management challenges, combined with inadequate nutritional knowledge and support, often resorted to restrictive dieting and unsustainable eating practices, further exacerbating weight regain. Another study emphasized that weight regain was significantly more common in individuals who lacked structured support for managing psychological and lifestyle factors, suggesting the importance of comprehensive postoperative care that addresses all three domains [18]. The findings from these studies indicate that a multidisciplinary approach to postoperative care is essential. Addressing dietary adherence alone is insufficient if psychological distress and sedentary lifestyles remain unaddressed. Instead, a tailored intervention plan incorporating dietary counseling, psychological support, and physical activity recommendations is likely to be more effective in promoting long-term weight maintenance and preventing weight regain.

## 4. Discussion

In this systematic review, we summarize the existing literature on dietary behaviors, psychological determinants, and lifestyle factors associated with weight regain following bariatric surgery. Our review focuses on key aspects such as dietary non-adherence, psychological influences, and lifestyle behaviors, identifying patterns that contribute to weight regain. Only 16 peer-reviewed articles met our inclusion and exclusion criteria, comprising 12 cross-sectional studies, 3 cohort studies, and 1 in-depth interview study. Although individual factors like dietary non-adherence, psychological determinants, and lifestyle behaviors have been recognized in the literature, this review uniquely integrates these factors to offer a comprehensive behavioral profile specific to post-bariatric surgery patients. Additionally, our analysis highlights inconsistencies in defining and measuring weight regain across studies, emphasizing the need for standardized assessment tools and definitions to improve comparability and accuracy in future research. Our findings reveal an underexplored intersection of psychological and dietary behaviors, suggesting that combined behavioral interventions may be essential for achieving sustainable weight management in this population.

### 4.1. Lack of Standardization in Weight Regain Criteria

Our review identified significant inconsistencies in defining weight regain after bariatric surgery, a challenge that is consistently noted across numerous studies [7,29,30,31,32]. Some studies use a decrease in excess weight loss, typically ranging from 10% to 25% of the lowest stable postoperative weight, as a measure of success [9,17]. However, this approach becomes less reliable as the likelihood of weight regain increases over time. The lack of a standardized definition complicates comparisons and the interpretation of results across studies.

Definitions varied considerably, with criteria based on the percentage of weight regained, BMI increase, or return to a baseline weight. This variability limits comparability and complicates conclusions on long-term outcomes. Based on our findings, we propose defining clinically significant weight regain as either a 15–20% increase from the patient’s lowest postoperative weight or a rise of 5 BMI units from their lowest achieved BMI. Standardizing this definition would improve the consistency of future research and provide healthcare providers with a reliable metric for monitoring postoperative weight management.

### 4.2. Dietary Non-Adherence and Eating Patterns

This review identified a variety of eating behaviors linked to weight regain, including emotional eating, loss of control eating, binge eating, and night eating. Many patients exhibited a lack of mindful eating practices, such as failing to dedicate at least 20 min to meals and a dependence on pre-prepared foods. Additional factors contributing to weight regain included cravings for sweets, increased consumption of breakfast foods, and disordered eating patterns, such as food addiction, problematic meal structuring, and large portion sizes. Patients also showed a notable preference for energy-dense foods and demonstrated gaps in nutritional knowledge. Behaviors like dietary restraint, increased fast food consumption, eating past fullness more than once a week, and generally poor diet quality were common, with some individuals frequently engaging in picking or nibbling throughout the day. This pattern, often involving small, intermittent portions rather than structured meals, can lead to unaccounted caloric intake, complicating weight management post-surgery. While not equivalent to binge eating, this behavior poses challenges for weight maintenance, as it frequently involves high-calorie, low-nutrient snacks in place of balanced meals.

Dietary non-adherence plays a significant role in weight regain, with participants often consuming a high intake of carbohydrates, fruits, and natural and canned juices, along with increased consumption of carbonated and energy drinks, contrasted with low vegetable intake and protein [23,33]. Post-surgery dietary restrictions in quantity may lead patients to seek calorie-dense foods. Several factors explain this shift. Firstly, the reduced stomach size after bariatric surgery limits the volume of food consumed at one time [34]. Secondly, high-fiber foods, such as fruits and vegetables, can cause digestive discomfort, including bloating and gas, in bariatric patients [35,36]. Thirdly, food intolerances may develop post-surgery, with certain fruits and vegetables becoming difficult to digest, leading to avoidance of these foods [37,38]. Lastly, changes in taste and food preferences are common, often reducing the desire to consume specific fruits or vegetables [39,40,41].

These findings suggest that eating patterns and behaviors after bariatric surgery can change over time, often toward less desirable patterns that some patients may not realize could impact their long-term health outcomes. It is evident that individuals who adopt such eating behaviors and subsequently experience weight regain may struggle with self-discipline and food control, an issue that could potentially exist even before undergoing surgery. Therefore, it is crucial to thoroughly assess such behaviors prior to the surgical procedure and recommend that the patient seek professional behavioral assistance for a significant period beforehand. Identifying these concerns before surgery imposes an ethical obligation to address them, as they could be the underlying cause of obesity, potentially rendering surgery unnecessary if behavioral intervention proves successful.

### 4.3. Psychological Determinants of Weight Regain Post-Bariatric Surgery

This study revealed several mental health factors associated with weight regain after bariatric surgery, including depression, anxiety, emotional eating, higher psychological distress, and unexpected weight management challenges. These psychological factors not only contribute directly to weight regain but also influence eating behaviors, often leading to maladaptive patterns such as emotional eating and binge eating. The complex relationship between mental health and eating behaviors highlights the need for comprehensive mental health support as part of post-surgical care, as addressing these factors is crucial for promoting sustainable weight management in bariatric patients.

Mental health conditions, particularly depression and anxiety, are strongly associated with maladaptive eating behaviors, including emotional eating, binge eating, and loss of control eating. Depression is often linked to alterations in appetite and eating patterns, with individuals experiencing depressive symptoms showing a tendency toward high-calorie, energy-dense food consumption as a form of self-soothing or emotional regulation [42]. Research suggests that depression can lead to increased cravings for carbohydrate-rich and sugary foods, as these can temporarily elevate mood through effects on brain neurotransmitters such as serotonin and dopamine [43]. However, this reliance on “comfort foods” often leads to weight gain, creating a feedback loop where weight-related self-esteem issues exacerbate depressive symptoms, further driving maladaptive eating behaviors [44].

Anxiety also plays a significant role in shaping eating behaviors. Individuals with anxiety disorders are more likely to experience heightened sensitivity to stress, which can trigger emotional eating or even binge eating episodes as a coping mechanism [45,46,47]. Anxiety-induced eating often occurs because eating provides temporary relief from psychological discomfort, distracting individuals from anxious thoughts and creating a sense of immediate but short-lived comfort [19,46,47]. In the context of anxiety, individuals may develop compulsive eating habits, feeling an urgency to consume food as a response to stress, regardless of hunger cues [46]. Over time, this pattern can lead to weight gain and additional anxiety about body image, reinforcing the cycle [46].

Emotional eating, a behavior common in both depression and anxiety, occurs when individuals eat in response to emotional states rather than physiological hunger. Studies show that individuals with high levels of depression and anxiety frequently turn to food as a way to manage negative emotions, using eating as an “escape” from distressing feelings [48,49]. This behavior is particularly problematic in individuals who have undergone bariatric surgery, as it can undermine their weight loss efforts and contribute to weight regain. A study indicated that emotional eating was closely associated with both depression and anxiety, with individuals reporting that food provided temporary relief from negative emotions but led to feelings of guilt and shame afterward, perpetuating a cycle of emotional distress and maladaptive eating [50].

In this systematic review, weight regain following bariatric surgery was associated with both depression and anxiety. The potential interaction between these two conditions remains unclear, presenting a gap in our understanding of causality and underlying mechanisms. However, to advance clinical practice, we recommend implementing routine screening for depression and anxiety both before and after bariatric surgery. This cost-effective approach could benefit patients by addressing mental health concerns that may contribute to weight regain, ultimately supporting both psychological well-being and long-term surgical success.

The interplay between mental health and eating behaviors highlights the importance of addressing psychological factors in weight management, particularly for individuals who have undergone bariatric surgery. Behavioral interventions, such as cognitive–behavioral therapy (CBT), have been effective in helping individuals recognize and modify the emotional triggers for eating, equipping them with strategies to break the cycle of emotional and binge eating. Integrating mental health support into weight management programs may be essential in addressing the root causes of maladaptive eating behaviors, leading to more sustainable outcomes for individuals struggling with both mental health issues and weight-related challenges.

### 4.4. Lifestyle Factors Contribute to Weight Regain

This review identified several lifestyle behaviors associated with weight regain after bariatric surgery, including physical inactivity, high levels of sedentary behavior, smoking, infrequent self-weighing, inadequate sleep, and limited support for weight management.

Physical inactivity and insufficient engagement in structured exercise routines are frequently linked to weight regain, with studies showing that regular physical activity helps maintain lean body mass, boosts metabolism, and reduces the likelihood of weight regain post-surgery [51,52,53,54,55]. High levels of sedentary behavior, including prolonged periods spent watching television, sitting, or using electronic devices, are also common in this population. Research indicates that sedentary behaviors are associated with metabolic slowing, insulin resistance, and increased fat accumulation, all of which contribute to weight regain over time [56].

Smoking, another prevalent factor, exacerbates health risks and weight maintenance challenges after bariatric surgery. Smoking is linked to increased metabolic complications, inflammation, and reduced physical fitness, making weight control more difficult and increasing the likelihood of complications [57,58,59]. Infrequent self-weighing was also noted, which diminishes self-monitoring and awareness of weight fluctuations. Studies show that individuals who self-weigh regularly are more likely to sustain their weight loss, as frequent monitoring fosters accountability, detects small weight gains early, and promotes timely adjustments to dietary and lifestyle habits [60,61].

Inadequate sleep, another factor commonly observed, disrupts metabolic processes, including glucose regulation and appetite control. Studies reveal that insufficient or poor-quality sleep increases levels of ghrelin (the hunger hormone) and decreases levels of leptin (the satiety hormone), which can lead to higher caloric intake and cravings for carbohydrate-rich foods [62]. Sleep deprivation is also associated with increased cortisol levels, which can promote fat storage and lead to weight regain [62].

Lastly, limited support for weight management, whether through social networks, professional guidance, or structured programs, further hinders sustained weight loss. Evidence shows that individuals with robust social and professional support networks experience greater success in adhering to dietary and exercise regimens, as support systems provide accountability, encouragement, and resources for managing weight challenges [63]. In the absence of such support, individuals may struggle to implement or maintain healthy behaviors, underscoring the need for comprehensive post-surgical support that addresses physical, emotional, and social dimensions of weight maintenance.

### 4.5. Complexities in Measuring Weight Regain and Its Determinants

Accurately measuring behaviors associated with weight regain presents inherent challenges, especially when relying solely on self-reported data, which can lead to both underestimations and overestimations. For instance, dietary behaviors may be underreported due to recall or social desirability biases, while physical activity levels are often overreported as individuals may exaggerate exercise routines. Furthermore, the lack of clarity regarding patients’ adherence to dietary guidelines or prescribed diet types complicates the assessment of dietary interventions’ effectiveness in preventing weight regain post-surgery. Without specific data on dietary protocols followed by patients, understanding the impact of dietary habits on long-term weight outcomes remains challenging.

Despite these measurement challenges, this review provides critical insights that healthcare specialists, nutritionists, psychologists, and support groups can utilize to improve weight maintenance strategies for bariatric surgery patients. Healthcare professionals can use these findings to identify key behavioral factors—such as dietary habits, psychological challenges, and lifestyle patterns—that contribute to weight regain. By addressing these factors early, they can develop personalized, targeted interventions tailored to specific patient needs. Nutritionists can apply the findings to create individualized dietary plans addressing common issues like grazing, emotional eating, and nutrient deficiencies. Educating patients on balanced meal planning, mindful eating, and managing cravings can improve adherence to a sustainable post-surgery diet, enhancing long-term weight maintenance.

Psychologists can leverage these insights to focus on mental health issues, such as depression, anxiety, and disordered eating behaviors, which are closely linked to weight regain. Therapeutic interventions, such as counseling and cognitive–behavioral therapy, can help patients develop healthier coping mechanisms and a more positive relationship with food, ultimately supporting improved emotional well-being. Support groups also play a pivotal role in reinforcing these behavioral changes by offering a platform for shared experiences, community support, and accountability. By discussing the behavioral challenges outlined in this review, support groups provide practical guidance and emotional support, further strengthening patients’ weight management efforts over time.

Given the variability in measurement tools across studies, there is a clear need for standardized and objective tools to assess physical activity, dietary adherence, and mental health in this population. Future research should aim to develop these tools and gather comprehensive data on participants’ adherence to specific dietary guidelines, along with any modifications made post-surgery. In-depth interviews with patients who experience weight regain could also provide valuable insights into the behavioral factors influencing long-term outcomes. Collectively, healthcare professionals and support systems can apply these insights within a multidisciplinary approach, enhancing the long-term success of bariatric surgery patients.

### 4.6. Limitation of This Review

The current review has several limitations that should be considered. First, there are relatively few studies examining behavioral factors associated with weight regain after bariatric surgery, which restricts the depth of available evidence. Additionally, the majority of included studies were cross-sectional with small sample sizes, limiting the ability to establish causal relationships between specific behaviors and weight regain. This highlights the need for future research that addresses the complexities of post-bariatric surgery weight management through more robust study designs. Moving forward, incorporating a broader range of methodologies, including longitudinal and larger-scale studies, could provide a more comprehensive understanding of the factors influencing long-term weight outcomes. Such research could inform the development of more effective interventions and support strategies tailored to the needs of bariatric surgery patients.

## 5. Conclusions

This systematic review highlights the complex interplay of dietary, psychological, and lifestyle factors contributing to weight regain after bariatric surgery. Key findings reveal that non-adherence to dietary guidelines, mental health challenges, and sedentary lifestyles significantly impact long-term weight outcomes. The variability in definitions and measures of weight regain across studies underscores the need for standardized criteria to enhance research comparability and improve clinical guidance. Additionally, the predominance of cross-sectional studies with small sample sizes limits our ability to draw causal inferences, pointing to a need for more rigorous, longitudinal research. For bariatric surgery patients to achieve sustainable weight management, a comprehensive, multidisciplinary approach is essential. Healthcare providers—including nutritionists, psychologists, and support groups—should address not only dietary adherence but also mental health and lifestyle behaviors. By integrating these insights, healthcare professionals can develop targeted, individualized interventions that address the unique needs of this population. Future research should aim to refine measurement tools and explore interventions that holistically address behavioral factors to optimize long-term success in weight maintenance after bariatric surgery.

## Figures and Tables

**Figure 1 healthcare-12-02243-f001:**
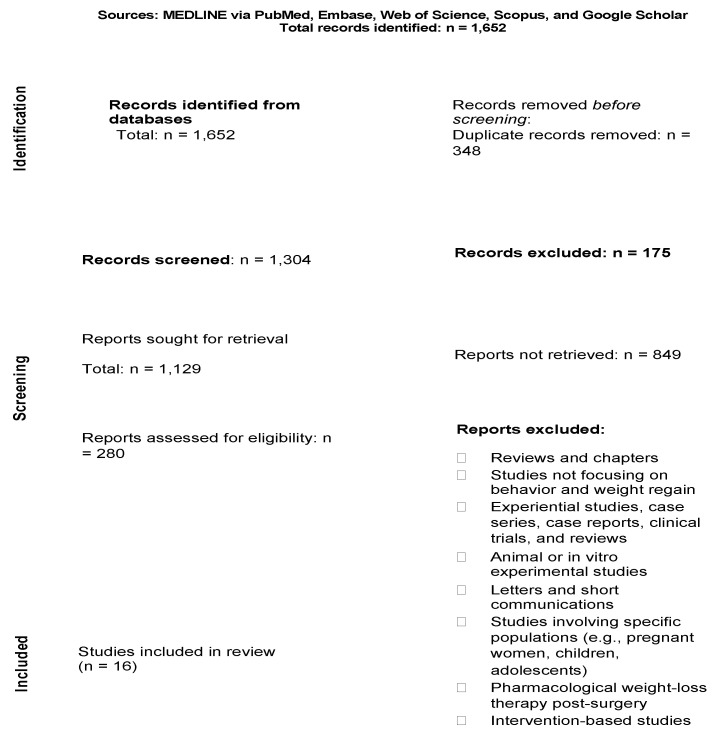
PRISMA 2020 flow diagram for new systematic reviews, which included searches of databases.

**Figure 2 healthcare-12-02243-f002:**
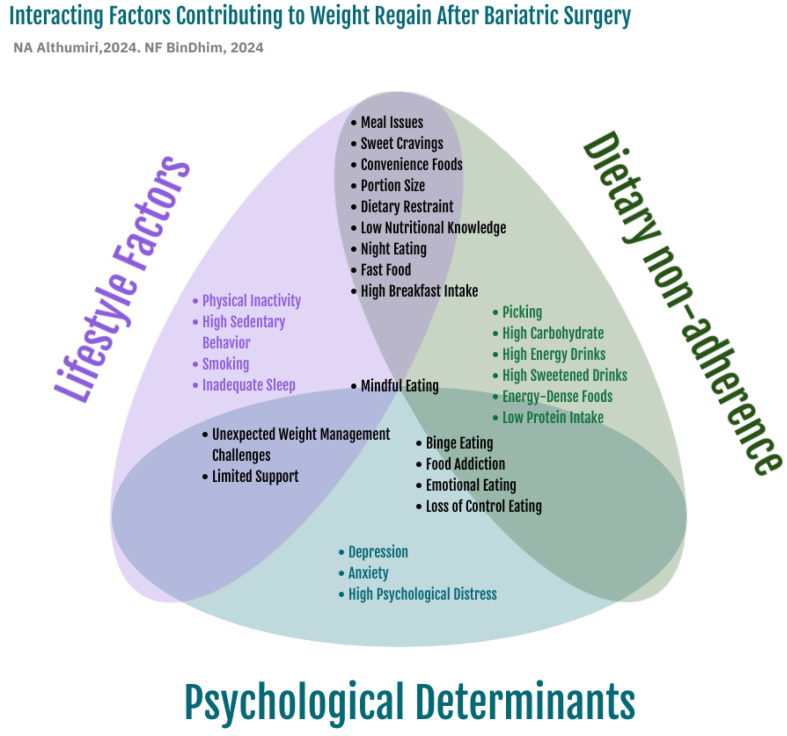
Illustrate interacting factors contributing to weight regain after bariatric surgery [14].

**Table 1 healthcare-12-02243-t001:** Summary of Included Studies on Behavioral Factors Associated with Weight Regain after Bariatric Surgery.

Reference and Year	Design/Time Since Surgery	Aims	Sample Size	Weight Gain	Summary of Results	Behavioral Factors Category
Althumiri, 2024 [15]	Cross-Sectional < 6 months	To explore the lifestyle attributes, psychological health aspects, and dietary patterns of individuals following bariatric surgery and compare their behavior to individuals with obesity who had not had any surgery.	806 participants	Weight gain was classified based on BMI, with 33% (n = 266).	Reclassified participants as “with obesity” after the surgery were associated with physical inactivity, smoking cigarettes, smoking waterpipes, efforts toward weight management, eating pre-prepared food (last week), eating breakfast meals, self-satisfaction rating, self-classification of weight, risk of anxiety, risk of depression, combined risk of anxiety and depression, fruit consumption level, vegetable consumption level, chicken consumption level, natural juice consumption level, canned juice consumption level, carbonated drink consumption, and energy drink consumption level.	Dietary non-adherence and eating patterns, psychological determinants, and lifestyle behaviors
Althumiri, 2023 [14]	Cross-Sectional < 6 M	To compare lifestyle and behavior between participants with and without obesity after bariatric surgery.	806 Participants	Weight gain was classified based on BMI, with 33% (n = 266).	Weight gain was associated with cigarette smoking, waterpipe smoking, self-managed weight control, self-classification of weight, and consumption of energy beverages, natural juices, and chicken.	Dietary non-adherence and eating patterns and lifestyle behaviors
Amundsen, 2018 [16]	Cross-Sectional ≥ 1 Y	To compare gastric bypass surgery patients experiencing suboptimal weight loss or significant weight regain with successful controls in terms of postoperative food intake, eating behavior, physical activity, and psychometrics.	49 Participants	Weight gain (≥15%) 77% (n = 38)	Weight gain was associated with the Three-Factor Eating Questionnaire, low physical activity, and reduced time spent in physical activity.	Dietary non-adherence and eating patterns and lifestyle behaviors
Berino, 2022 [17]	Cross-Sectional > 2 Y and <10 Y	To investigate the influence of Quality of Life (QOL) on weight regain.	50 women	Weight gain (≥15%) 60% (n = 30)	Weight gain was positively associated with uncontrolled eating and the physical component of Quality of Life (QOL).	Dietary non-adherence and eating patterns and lifestyle behaviors
Conceição, 2014 [18]	Cross-Sectional (6 m, 1 y, 2 y)	To describe the presence of various maladaptive eating behaviors—objective binge eating (OBE), subjective binge eating (SBE), and picking or nibbling (P&N)—at pre-surgery and at follow-up points ranging from short-term to long-term.	321 Participants	Weight gain at 2 Y (LAGB = 25.4) and (LRYGB = 19.2)	Weight gain was associated with picking or nibbling and higher psychological distress.	Dietary non-adherence and eating patterns and psychological determinants
Da Silva, 2016 [12]	Cohort Study > 2 Y	To investigate factors associated with long-term weight regain after Roux-en-Y gastric bypass.	80 Participants	weight gain (10%) 23.7%	Weight gain was associated with poor diet quality as measured by the Healthy Eating Index.	Dietary non-adherence and eating patterns
Dos Rodrigues, 2021 [9]	Cross-Sectional > 2 Y	To study the association between weight gain and physical activity.	44 women	Weight gain (≥15%) 59,1% (n = 26)	Weight gain was inversely associated with eating restriction behaviors and time spent on physical activity.	Dietary non-adherence and eating patterns and lifestyle behaviors
Freire, 2021 [19]	Cross-Sectional 1. 2 Y; 2.7–14 Y	To assess weight gain and the long-term recurrence of binge eating, depressive symptoms, and anxious symptoms at three periods: preoperative, 24 months post-operative, and long-term	96 patients at 2 Y, and 46 patients at 7–14 Y.	Weight Gain (≥20%) 1. 8% (n = 96) 2. 67.39% (n = 31)	At the 2-year follow-up, binge eating, depression, and anxiety decreased; however, at the 7–14-year follow-up, weight gain was associated with binge eating. A higher percentage of weight gain was observed among those with depressive and anxiety symptoms.	Dietary non-adherence and eating patterns and psychological determinants
King, 2020 [13]	Cohort Study	To assess patient behavior that influenced weight gain.	1278 Patients	The median weight gain was 25.2% (25–75th percentile, 14.0–39.3%) of the maximum weight lost.	Weight gain related to behavior was associated with greater sedentary time, eating more fast food, eating when full more than once a week, continuous eating throughout the day, binge eating, night eating, loss of control eating, less frequent self-weighing, and higher depressive symptoms.	Dietary non-adherence and eating patterns, psychological determinants, and lifestyle behaviors
McInnis, 2022 [20]	Cross-Sectional > 2 Y.	To investigate appetite-related factors associated with weight regain after Roux-en-Y Gastric Bypass (RYGB) surgery.	29 Participants	1. Low weight gain 10.0 ± 3.4 kg; 2. High weight gain 14.9 ± 6.3 kg	Dietary restraint was significantly higher in both weight gain groups.	Dietary non-adherence and eating patterns
Miller-Matero, 2024 [21]	Cross-Sectional > 2 Y.	To examine whether psychiatric symptoms, maladaptive eating behaviors, and lifestyle factors were associated with weight recurrence.	169 Participants	Weight gain 23.1% (39 Participants)	Weight gain was associated with anxiety and depressive symptoms, emotional eating, loss of control eating, binge eating, and night eating. It was also linked to individuals who did not eat mindfully, did not take 20 min to eat, or did not get adequate sleep.	Dietary non-adherence and eating patterns lifestyle behaviors
Monpellier, 2019 [22]	Cohort Study at 15, 24, 36 and 48 M.	To assess the relationship between weight change, self-reported physical activity, and eating style.	4569 Patients	The mean weight gain at 36 months was 5.3% ± 6.7, and at 48 months, it was 7.2% ± 9.2.	Higher weight gain at 36 months was associated with restrained eating. Higher weight regain at 48 months was positively associated with emotional eating. Physical activity was negatively associated with weight gain. Patients who exhibited more emotional eating and external eating were more likely to experience weight gain.	Dietary non-adherence and eating patterns and psychological determinants
Nicanor-Carreón, 2023 [23]	Cross-Sectional 2–10 Y.	To investigate maladaptive eating behaviors and weight regain in individuals who are women.	36 women	Weight gain (≥20%) 44.4% (n = 16)	Weight gain was positively associated with the frequency of cravings for sweets, measures of eating disorders, food addiction, and loss of control when eating.	Dietary non-adherence and eating patterns
Romagna, 2021 [24]	Cross-Sectional 1. <5 Y; 2. >5 Y	To investigate physical activity levels, sedentary time (ST), and weight regain in patients who did not have regular medical follow-up prior to recruitment.	90 Participants	weight gain (>20) = 44% (40)	Weight gain was greater among individuals with low levels of physical activity and a more sedentary lifestyle.	Lifestyle behaviors
Tolvanen, 2023 [25]	In-depth, semi-structured individual interviews	To explore patients’ perspectives on dietary patterns and eating behaviors during weight regain after bariatric surgery.	16 participants	-	Weight gain was linked to higher perceptions of dietary challenges, such as unexpected weight management issues, problematic meal patterns, increasing portion sizes, and appealing energy-dense foods. Disordered eating patterns and emotional eating. Insufficient nutritional knowledge and a lack of support further hindered participants’ ability to avoid weight regain.	Dietary non-adherence and eating patterns, psychological determinants, and lifestyle behaviors
Vieira, 2019 [26]	Cross-Sectional > 2 Y	To investigate the perception of hunger and satiety and its association with nutrient intake among women experiencing weight regain.	60 women	Weight gain (10%) 33.3% (n = 20)	Weight gain was associated with lower protein intake and negatively with carbohydrate intake.	Dietary non-adherence and eating patterns

Note: “Behavioral Factors Category” refers to the main thematic findings, including dietary non-adherence, psychological determinants, and lifestyle behaviors, as discussed in the Results Section.

## Data Availability

Data extracted from the included studies and used for all analyses are available upon request. Please contact na@idm.net to obtain access to the data.

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
