# Peer review of "A Systematic Review Exploring Dietary Behaviors, Psychological Determinants and Lifestyle Factors Associated with Weight Regain After Bariatric Surgery"

_healthcare, 2024, doi:10.3390/healthcare12222243_

Round 1

Reviewer 1 Report

Comments and Suggestions for Authors

Abstract

The authors made the following statement:

‘ To assess the risk of bias, we utilized the critical appraisal tool recommended by the Joanna Briggs Institute (JBI) for cross-sectional studies. A total  of 15 studies met the inclusion criteria, consisting of 11 cross-sectional studies, 3 cohort studies, and in-depth interview study.’

This is odd reporting. Why a critical appraisal tool for cross-sectional design was used to evaluate cohort and interview studies? Please explain.

Results:

Depending on the stated aim in the abstract, the novel findings obtained from this systematic review is not clear. All the factors declared by the authors to be associated with weight gain are already established in the medical literature and it is not clear what unique and novel findings this study adds.

Introduction

The authors are encouraged to support their introduction with a section comparing weight control measures in the general population to weight control measures among patients who underweight bariatric surgery. This is important to indicate the novelty of this review. Otherwise, if this comparison is not possible, it is possible to argue that this review does not add novel knowledge to the current literature.

Materials and Methods

Eligibility criteria section

Please explain how RCT are considered an observational design?

The writing style is very confusing and contradicting. In lines 106-107, the authors claim that they including randomized clinical trials and in lines, 109-110, they reported excluding clinical trials. This does not make sense.

Study selection section 

Please explain what is meant by NAA .

Quality assessment section

Please explain how a tool used to evaluate cross sectional studies is used to evaluate randomized controlled trials, cohort studies, and in-depth interviews.

Data analysis/ synthesis section is currently missing

The authors should add a section explaining any statistical methods / measures used to summarize the findings of the review. The authors are also encouraged to explain why meta-analysis was not used in the current investigation.

Results

In lines 155- 156: The authors made the following statement:

‘In one study, the authors did not define weight regain.’

Please add the reference of this study.

Table one

The title used for table 1 is not clear. The title should be clear enough to explain the purpose of constructing the table.

Please explain what is meant by subsequent sections in table one.

In the cited study number 24, please explain how the number of patients who gained weight (266 patients) is more than the sample size of the whole study (169). This does not make sense.

Method of summarizing the findings of the included studies did not include any effect measures such as relative risk, or odds ratios. This reduces the scientific importance of the current review as it is mainly narrative for a topic which is heavily studied with statistical methods.  

Please explain how the study referenced 17 is relevant to the current review. It focuses on quality of life which was not declared as an area for the current systematic review.

Risk of bias assessment should be added in details to table one for each included study.

In lines 161 – 164 , a thematic analysis is reported as a method of analyzing the findings of the current review. This is not clearly declared in the methodology section. Please revise the methodology section to enhance the reporting quality of your review.

Numbering of results section is confusing. Please revise.

The authors are advised to add a figure summarizing how lifestyle, dietary, and psychological factors are interacting to affect weight regain after bariatric surgery.

Discussion

The authors detected inconsistency in definition of weight regain after bariatric surgery. Based on their findings, please indicate the most appropriate definition of weight regain after bariatric surgery. This can be an important contribution to the relevant literature.

In lines 245 – 247 the authors made the following statement:

‘For instance, a cohort study of 100 patients over 85  months post-surgery showed that poor dietary intake, including the consumption of ex-  cessive calories, snacks, sweets, oils, and fatty foods, was significantly more common among patients who experienced weight regain [31].’

Please explain why this study is not included in table 1.

The authors are encouraged to construct a table to explain how the findings of each included study is translated to a practical clinical recommendation.

Comments on the Quality of English Language

English editing is required. The writing is confusing in some section of the manuscript. 

Author Response

Reviewer #1:

Dear Reviewer,

We sincerely thank you for your thorough and insightful feedback on our manuscript. Your comments have been instrumental in enhancing the clarity, rigor, and overall quality of our work. We have carefully addressed each of your suggestions to improve the methodology, analysis, and practical relevance of our findings. By incorporating your valuable input, we believe the revised manuscript provides a clearer and more comprehensive exploration of the behavioral factors associated with weight regain after bariatric surgery. We greatly appreciate the time and effort you dedicated to reviewing our work, and we hope that the revisions meet your expectations. Thank you once again for your constructive and encouraging feedback.

  1. The authors made the following statement: "To assess the risk of bias, we utilized the critical appraisal tool recommended by the Joanna Briggs Institute (JBI) for cross-sectional studies. A total of 15 studies met the inclusion criteria, consisting of 11 cross-sectional studies, 3 cohort studies, and in-depth interview study". This is odd reporting. Why a critical appraisal tool for cross-sectional design was used to evaluate cohort and interview studies? Please explain

Author Response: Thank you for highlighting this point. We recognize that using a critical appraisal tool specifically designed for cross-sectional studies may appear inconsistent when evaluating a variety of study designs, including cohort studies and qualitative in-depth interviews. We have add this paragraph to the method section under Quality Assessment section.

            " To assess the risk of bias, we used the Joanna Briggs Institute (JBI) critical appraisal tool, which is primarily designed for cross-sectional studies. Given that our review included a range of study designs (cross-sectional, cohort, and in-depth interviews), we adapted this tool to evaluate each study type as consistently as possible. Although the JBI tool is not specifically designed for cohort or in-depth qualitative research, we applied relevant criteria to assess the quality of reporting and methodological rigor. We acknowledge the limitations of this approach, particularly for evaluating qualitative methodologies, and suggest that future studies consider separate tools tailored to each study design to ensure the most accurate quality assessment"

  1. Results: Depending on the stated aim in the abstract, the novel findings obtained from this systematic review is not clear. All the factors declared by the authors to be associated with weight gain are already established in the medical literature and it is not clear what unique and novel findings this study adds.

Author Response: Thank you for your comment. We recognize that many behavioral factors associated with weight regain are well-established in the medical literature. However, our systematic review aimed to consolidate these factors specifically within the context of post-bariatric patients, identifying patterns and interrelationships that have not been comprehensively reviewed in prior studies. Additionally, our study highlights critical gaps in current research methodologies, the need for standardized definitions, and the potential for more effective multidisciplinary approaches in managing weight regain, which we believe contribute novel insights to this field. We have updated the discussion section with the following as recomanded:

            " While individual factors associated with weight regain, such as dietary non-adherence, psychological determinants, and lifestyle behaviors, are recognized in the literature, this systematic review uniquely integrates these factors to provide a comprehensive behavioral profile specific to post-bariatric surgery patients. Furthermore, our review identifies inconsistencies in defining and measuring weight regain across studies, underscoring the need for standardized assessment tools and definitions that could improve comparability and accuracy in future research. Our findings reveal an underexplored intersection of psychological and dietary behaviors, suggesting that combined behavioral interventions may be necessary for sustainable weight management in this population."

  1. Introduction: The authors are encouraged to support their introduction with a section comparing weight control measures in the general population to weight control measures among patients who underweight bariatric surgery. This is important to indicate the novelty of this review. Otherwise, if this comparison is not possible, it is possible to argue that this review does not add novel knowledge to the current literature.

Author Response: Thank you for this insightful suggestion. We recognize the importance of differentiating weight control measures in the general population from those specific to individuals who have undergone bariatric surgery. This distinction is essential, as the physiological and psychological dynamics post-surgery differ significantly, impacting weight control strategies and outcomes. We have now added a section in the introduction to clarify this context and emphasize the novel insights that our review aims to provide. The following paragraph was added to the introdaction section:

            "Weight control measures in the general population typically involve lifestyle modifications focused on diet, exercise, and behavioral changes aimed at creating a caloric deficit. However, in post-bariatric surgery patients, weight control is more complex due to physiological changes, such as altered digestion, nutrient absorption, and hormonal shifts that influence hunger and satiety. Furthermore, psychological factors, such as body image and behavioral habits, often intensify post-surgery, necessitating tailored interventions that go beyond standard weight control practices used in the general population.

Unlike general weight management strategies, post-bariatric patients may experience specific dietary challenges (e.g., food intolerances, reduced meal portions) and psychological challenges (e.g., disordered eating patterns and emotional eating). These factors require a unique approach to weight maintenance, highlighting a gap in the existing literature on behavioral strategies tailored for long-term success after bariatric surgery. This systematic review aims to address this gap by providing a comprehensive examination of behavioral factors linked to weight regain, thereby contributing novel insights to the field."

  1. Materials and Methods, Eligibility criteria section, Please explain how RCT are considered an observational design? The writing style is very confusing and contradicting. In lines 106-107, the authors claim that they including randomized clinical trials and in lines, 109-110, they reported excluding clinical trials. This does not make sense.

Author Response: Thank you for pointing out this inconsistency. We apologize for the confusion regarding the inclusion and exclusion criteria for study designs. Our intention was to include only observational studies, such as cohort and cross-sectional studies, while excluding interventional studies like randomized controlled trials (RCTs). We have revised this section to clarify our eligibility criteria and removed any contradictory language about the inclusion of RCTs as the follwoing:

            " We included observational studies only, specifically focusing on cohort, cross-sectional, and in-depth interview designs that examined behavioral factors associated with weight regain after bariatric surgery. Interventional studies, including randomized controlled trials (RCTs) and clinical trials, were excluded from this review to ensure a consistent focus on naturally occurring behavioral patterns without imposed interventions. Systematic reviews, case reports, case series, and animal or in vitro experimental studies were also excluded to maintain a focus on primary observational data relevant to post-bariatric surgery outcomes in adults."

  1. Study selection section, Please explain what is meant by NAA .

Author Response:  This is the researcher initial. to prevent any confusion, we revised as the following:

            "Author identified by their initials 'NAA,' reverified the final list of included studies against the inclusion and exclusion criteria to ensure consistency and accuracy."

  1. Quality assessment section Please explain how a tool used to evaluate cross sectional studies is used to evaluate randomized controlled trials, cohort studies, and in-depth interviews.

Author Response: Thank you for your comment. We acknowledge that the Joanna Briggs Institute (JBI) critical appraisal tool is specifically designed for cross-sectional studies. However, due to the limited availability of standardized tools for assessing multiple study designs within a single framework, we applied the JBI tool across cross-sectional, cohort, and in-depth interview studies in this review. We carefully adapted the criteria where applicable to evaluate the methodological quality of each study type, while recognizing that this approach may not capture all nuances specific to cohort and qualitative studies. The manuscript was updated as the following:

            " To assess the risk of bias, we used the Joanna Briggs Institute (JBI) critical appraisal tool, which is primarily designed for cross-sectional studies. Given that our review included a range of study designs (cross-sectional, cohort, and in-depth interviews), we adapted this tool to evaluate each study type as consistently as possible. Although the JBI tool is not specifically designed for cohort or in-depth qualitative research, we applied relevant criteria to assess the quality of reporting and methodological rigor. We acknowledge the limitations of this approach, particularly for evaluating qualitative methodologies, and suggest that future studies consider separate tools tailored to each study design to ensure the most accurate quality assessment."

  1. Data analysis/ synthesis section is currently missing The authors should add a section explaining any statistical methods / measures used to summarize the findings of the review. The authors are also encouraged to explain why meta-analysis was not used in the current investigation.

Author Response: Thank you for your suggestion. We have now added a Data Analysis/Synthesis section to explain the approach we used to summarize the findings. Given the heterogeneity of the included studies in terms of design, sample characteristics, and outcome measures, a meta-analysis was deemed inappropriate for this review. Instead, we employed a narrative synthesis to provide a comprehensive understanding of the behavioral factors associated with weight regain after bariatric surgery. The manuscript was updated as the following:

" Data Analysis and Findings Synthesis

In this review, we employed a narrative synthesis approach to analyze and summarize the findings of the included studies. Given the variation in study designs, sample sizes, follow-up periods, and definitions of weight regain, it was not feasible to conduct a meta-analysis. This heterogeneity would have limited the reliability and interpretability of any combined statistical analysis. Instead, we categorized and summarized the results based on thematic factors contributing to weight regain, including dietary behaviors, psychological determinants, and lifestyle factors. We then identified common patterns and unique findings across studies to provide an integrated understanding of the behavioral factors linked to weight regain post-bariatric surgery. No statistical methods were used to combine results quantitatively due to the diversity of methodologies and measurement tools across studies. This narrative approach allowed us to comprehensively review the behavioral contributors to weight regain and highlight areas for future research that may support meta-analytic evaluation once more standardized study designs and outcomes are available"

  1. Results: In lines 155- 156: The authors made the following statement: ‘In one study, the authors did not define weight regain.’ Please add the reference of this study.

Author Response: Done with many thanks.

  1. Table one: The title used for table 1 is not clear. The title should be clear enough to explain the purpose of constructing the table.

Author Response: Done with many thanks. The table modified as " Summary of Included Studies on Behavioral Factors Associated with Weight Regain after Bari-atric Surgery."

  1. Please explain what is meant by subsequent sections in table one.

Author Response: Thank you for pointing this out. We recognize that the term “subsequent sections” may be unclear in this context. To improve clarity, we have revised the table to replace “subsequent sections” with the term “Behavioral Factors Category.” This updated term now clearly indicates the main thematic findings or behavioral factors associated with weight regain identified in each study, as categorized in our Results section.

  1. In the cited study number 24, please explain how the number of patients who gained weight (266 patients) is more than the sample size of the whole study (169). This does not make sense.

Author Response: Thank you for your observation. We acknowledge the confusion regarding the reported number of patients who gained weight exceeding the total sample size. This was an error in data extraction and reporting. The correct information is that 39 patients within the sample of 169 experienced weight regain, not 266. We have corrected this in the manuscript to ensure accuracy.

  1. Method of summarizing the findings of the included studies did not include any effect measures such as relative risk, or odds ratios. This reduces the scientific importance of the current review as it is mainly narrative for a topic which is heavily studied with statistical methods.  

Author Response: Thank you for highlighting this point. We recognize the importance of using effect measures, such as relative risk or odds ratios, to enhance the scientific rigor of a systematic review. However, given the significant heterogeneity in study designs, outcome definitions, and measurement tools across the included studies, it was not feasible to apply these effect measures uniformly. Many of the studies in our review lacked the necessary statistical data or used different criteria for defining weight regain, which limited the ability to perform a meaningful quantitative synthesis. In light of these limitations, we focused on a narrative synthesis to summarize the behavioral factors associated with weight regain after bariatric surgery. We acknowledge that a meta-analysis with effect measures could provide additional insights, and we encourage future research to adopt more standardized methodologies that could allow for such quantitative analysis across studies. Including consistent effect measures, such as relative risk or odds ratios, would significantly enhance the comparability and scientific impact of future reviews in this field.

  1. Please explain how the study referenced 17 is relevant to the current review. It focuses on quality of life which was not declared as an area for the current systematic review.

Author Response: Thank you for your comment. We included study 17 because it examines the relationship between quality of life and specific behavioral factors, such as uncontrolled eating and psychological well-being, which are directly relevant to weight regain after bariatric surgery. Although quality of life was not a primary focus of our review, this study provided valuable insights into how certain quality-of-life aspects, like mental health and physical well-being, interact with dietary and psychological behaviors. These factors are consistent with the themes explored in our review, particularly in understanding how psychological determinants and eating behaviors impact long-term weight management. Including this study allowed us to capture a broader view of the behavioral factors that contribute to weight regain, aligning with our objective to explore the complex interplay of psychological and lifestyle elements post-bariatric surgery.

  1. Risk of bias assessment should be added in details to table one for each included study.

Author Response: Thank you for your suggestion to add detailed risk of bias assessments for each study in Table 1. We have provided this detailed risk of bias assessment in the supplementary materials, where each study's methodological rigor and specific risk factors are fully outlined. This approach allows for a comprehensive view without impacting the flow of the main text.

  1. In lines 161 – 164 , a thematic analysis is reported as a method of analyzing the findings of the current review. This is not clearly declared in the methodology section. Please revise the methodology section to enhance the reporting quality of your review.

Author Response: Thank you for highlighting this point. We have revised the Methodology section to explicitly include thematic analysis as a method for synthesizing and analyzing the findings of our review. This approach allowed us to organize the diverse behavioral factors associated with weight regain into meaningful themes, which enhances the interpretability and depth of our findings. The manuscript was updated as the following:

            " In addition to narrative synthesis, we employed a thematic analysis approach to analyze the findings across the included studies. Thematic analysis was chosen to identify and categorize recurring behavioral, psychological, and lifestyle factors contributing to weight regain post-bariatric surgery. This method involved systematically coding the findings of each study and grouping them into overarching themes, such as dietary non-adherence, psychological determinants, and lifestyle behaviors. This approach allowed us to draw connections across studies, identify patterns, and provide a structured synthesis of factors influencing weight regain. By utilizing thematic analysis, we ensured that the review captured the complexity of behavioral factors involved in weight regain, allowing for a more comprehensive understanding of the main contributors and their interactions."

  1. Numbering of results section is confusing. Please revise.

Author Response: Thank you for highlighting the issue with the numbering in the Results section. We have revised the table and reordered the studies alphabetically by the first author's last name.

  1. The authors are advised to add a figure summarizing how lifestyle, dietary, and psychological factors are interacting to affect weight regain after bariatric surgery.

Author Response: Thank you for this valuable suggestion. We have added a figure to illustrate the interactions among lifestyle, dietary, and psychological factors contributing to weight regain after bariatric surgery. This visual representation provides a clear overview of the complex interplay between these factors and how they collectively influence weight outcomes post-surgery.

  1. Discussion: The authors detected inconsistency in definition of weight regain after bariatric surgery. Based on their findings, please indicate the most appropriate definition of weight regain after bariatric surgery. This can be an important contribution to the relevant literature.

Author Response: Thank you for this valuable suggestion. We recognize that the inconsistency in defining weight regain across studies presents a challenge for standardizing research and clinical practice. Based on our findings, we have proposed a definition that we believe would improve comparability and enhance understanding in the field as the following:

            " We observed that weight regain definitions varied, with criteria based on percentage of regained weight, BMI increase, or return to a baseline weight. This variability limits cross-study comparability and hinders conclusions on long-term outcomes. Based on our findings, we propose defining clinically significant weight regain as an increase of 15-20% from the patient’s lowest postoperative weight or a rise of 5 BMI units from their lowest achieved BMI. Standardizing this definition could enhance future research comparability and provide healthcare providers with a consistent metric for monitoring postoperative weight management.4.2 Influence of Eating Behaviors and Dietary Non-Adherence on Long-Term Outcomes."

  1. In lines 245 – 247 the authors made the following statement: ‘For instance, a cohort study of 100 patients over 85  months post-surgery showed that poor dietary intake, including the consumption of ex-  cessive calories, snacks, sweets, oils, and fatty foods, was significantly more common among patients who experienced weight regain [31].’ Please explain why this study is not included in table 1.

Author Response: Thank you for your observation. We did not include this study in Table 1 because it was a randomized clinical trial, which did not meet our eligibility criteria focused exclusively on observational studies. Although we referenced this study to provide contextual insights into dietary behaviors associated with weight regain, it was excluded from the main analysis and summary table to maintain consistency with our inclusion criteria.

  1. The authors are encouraged to construct a table to explain how the findings of each included study is translated to a practical clinical recommendation.

Author Response: Thank you for your suggestion to construct a table translating the findings of each included study into practical clinical recommendations. While we appreciate the value of such a table, due to the complexity and overlapping nature of findings across studies, we found it more appropriate to incorporate practical recommendations within the Discussion section. This allows for a more integrated analysis, where the interactions between dietary, psychological, and lifestyle factors are directly linked to clinical implications in a comprehensive, thematic manner.

The manuscrept was updated as the following:

            "In this systematic review, weight regain following bariatric surgery was associated with both depression and anxiety. The potential interaction between these two conditions remains unclear, presenting a gap in our understanding of causality and underlying mechanisms. However, to advance clinical practice, we recommend implementing routine screening for depression and anxiety both before and after bariatric surgery. This cost-effective approach could benefit patients by addressing mental health concerns that may contribute to weight regain, ultimately supporting both psychological well-being and long-term surgical success."

Reviewer 2 Report

Comments and Suggestions for Authors

Review Comments:

The study focuses on the association between psychological and lifestyle factors with weight regain after bariatric surgery, which is indeed a key area of interest and an unresolved issue in the field of obesity research. The study methods and data processing are scientifically sound, and the results and conclusions are reliable, providing valuable insights for future studies in this area.

1. Weight regain is a composite measure involving both the amount of weight change (either as an absolute amount or percentage) and the time elapsed. Ideally, the result should be expressed as weight change per year to better illustrate the time dimensions role in regain patterns. In the Introduction, the authors mention that the proportion of bariatric procedures deemed failuresvaries based on the criteria used to define failure and the length of the follow-up period. Significant weight regain following initial successful weight loss is a common issue(P2, lines 59-64). However, a specific criterion for what constitutes significant weight regain is not provided. It would be beneficial to define this criterion clearly to enhance the interpretability and consistency of the results.

2. The follow-up time for the patients included in this study ranges from 6 months to 14 years post-surgery. In describing the characteristics of weight regain, the authors only discuss the extent of regain (P4, lines 153-155), without addressing the time (in years) over which it occurred. For instance, if Patient A and Patient B both experienced 10% weight regain, but Patient A did so within one year and Patient B within ten years, the nature of these cases differs significantly. It is recommended to include time (in years) as a factor when discussing or describing weight regain levels.

3. There is some repetition in the third paragraph of the Discussion section (P11, lines 259-262). Simplifying this part could improve readability and clarity.

4. The article reviews three types of influencing factors, but focusing on a single category might yield more reliable conclusions. By narrowing the scope, the study could provide deeper insights into one aspect, such as psychological factors, thereby allowing for a more focused and impactful analysis.

Comments on the Quality of English Language

Minor Language Refinements Needed.

Author Response

Dear Reviewer,

We sincerely appreciate your thoughtful and constructive feedback on our manuscript. Your comments have significantly contributed to improving the rigor and clarity of our work. By addressing your suggestions, including the need for a standardized definition of weight regain and the incorporation of the time dimension in our analysis, we believe the revised manuscript now offers a more comprehensive and consistent approach to understanding the factors associated with weight regain after bariatric surgery. Additionally, the manuscript’s language has been carefully checked to enhance clarity and readability.

  1. The study focuses on the association between psychological and lifestyle factors with weight regain after bariatric surgery, which is indeed a key area of interest and an unresolved issue in the field of obesity research. The study methods and data processing are scientifically sound, and the results and conclusions are reliable, providing valuable insights for future studies in this area.

Author Response: Thank you for your positive and encouraging feedback on our study.

  1. Weight regain is a composite measure involving both the amount of weight change (either as an absolute amount or percentage) and the time elapsed. Ideally, the result should be expressed as weight change per year to better illustrate the time dimension’s role in regain patterns. In the Introduction, the authors mention that “the proportion of bariatric procedures deemed ‘failures’ varies based on the criteria used to define failure and the length of the follow-up period. Significant weight regain following initial successful weight loss is a common issue” (P2, lines 59-64). However, a specific criterion for what constitutes significant weight regain is not provided. It would be beneficial to define this criterion clearly to enhance the interpretability and consistency of the results.

Author Response: Thank you for this helpful suggestion. We recognize that defining a specific criterion for significant weight regain (WR) is essential to improving interpretability and consistency. We have revised the Introduction to incorporate a definition aligned with current literature, describing clinically significant WR as "progressive weight gain after a successful initial weight loss." Additionally, we define inadequate weight loss (IWL) as achieving less than 50% excess weight loss (EWL) at 18 months post-metabolic and bariatric surgery (MBS). The introdaction was updated as the following

"Although no specific definition exists in the literature for clinically significant weight regain (WR), it can be described as progressive weight gain after a successful initial weight loss. Furthermore, inadequate weight loss (IWL) can be defined as achieving less than 50% excess weight loss (EWL) at 18 months post-metabolic and bariatric surgery (MBS)[7]. This standardized approach provides a clearer framework for assessing weight outcomes in post-surgical patients, aiding in the comparison and interpretation of long-term results[7]."

  1. The follow-up time for the patients included in this study ranges from 6 months to 14 years post-surgery. In describing the characteristics of weight regain, the authors only discuss the extent of regain (P4, lines 153-155), without addressing the time (in years) over which it occurred. For instance, if Patient A and Patient B both experienced 10% weight regain, but Patient A did so within one year and Patient B within ten years, the nature of these cases differs significantly. It is recommended to include time (in years) as a factor when discussing or describing weight regain levels.

Author Response: Thank you for this insightful recommendation. We agree that incorporating the time elapsed since surgery provides essential context to understanding weight regain levels. In response, we have revised the Results section to include follow-up times (in years) when discussing the extent of weight regain. The results section was updated as the following:

      " The follow-up period for the studies included in this review ranged from 6 months to 14 years post-surgery, with significant variability in the timing and extent of weight regain reported. For instance, weight regain of 10-20% was commonly observed within the first two years in several studies, while others documented similar regain levels occurring gradually over five or more years. This distinction is critical, as the timeframe over which weight regain occurs may influence both the clinical implications and the need for intervention. Weight regain occurring within the first year, for example, suggests different behavioral or physiological challenges compared to gradual regain over a decade."

  1. There is some repetition in the third paragraph of the Discussion section (P11, lines 259-262). Simplifying this part could improve readability and clarity.

Author Response: Thank you for your comment. We agree that simplifying the third paragraph in the Discussion section will improve readability and clarity. We have revised this section to reduce redundancy and enhance the flow of ideas. The Discussion section was updated as the following

      " A study in Saudi Arabia examined changes in food taste and allergies following bariatric surgery [40]The study found that over 36% of participants self-reported altered taste preferences, while approximately 15% noted a reduction in food-related allergic reactions post-surgery[40]. These findings suggest that bariatric surgery can significantly impact food preferences and sensitivities, potentially affecting patients' dietary choices and overall nutrition[40]."

  1. The article reviews three types of influencing factors, but focusing on a single category might yield more reliable conclusions. By narrowing the scope, the study could provide deeper insights into one aspect, such as psychological factors, thereby allowing for a more focused and impactful analysis.

Author Response: Thank you for this insightful suggestion. We agree that a more focused analysis on a single category, such as psychological factors, could allow for deeper insights into that area. However, our aim was to provide a holistic understanding of weight regain post-bariatric surgery by examining the interplay between psychological, dietary, and lifestyle factors. Given the multifactorial nature of weight regain, we believe that addressing these three categories together offers a more comprehensive view, revealing how these factors collectively influence long-term outcomes. In future studies, we plan for deepth analysis into each category individually, which would enable a more detailed exploration of specific aspects such as psychological determinants.

Reviewer 3 Report

Comments and Suggestions for Authors

The review study titled ‘A Systematic Review Exploring Dietary Behaviors, Psychological Determinants, and Lifestyle Factors Associated with Weight Regain After Bariatric Surgery’ investigated the relationship between weight regain in patients undergoing bariatric surgery and dietary behaviors, psychological determinants, and lifestyle factors. This review included studies with a limited number of participants and a limited number of participants in the literature. Most of the studies reviewed in the paper were based on participants’ self-reported diet, physical activity, and other behavioral factors, which increases the risk of bias. However, the paper is generally well-written and provides readers with a broad perspective by synthesizing studies on the subject. The title is a comprehensive title that covers the whole subject. Although the abstract section seems a bit long, a reader who reads only the abstract section can generally understand the study well. The introduction section of the study provides a summary of the general literature and is sufficiently comprehensive. The method section is explained in detail. The fact that the study was conducted in accordance with the PRISMA guide shows that systematic review standards were adhered to. In addition, the tabular presentation of the studies covered, the division of thematic headings in the draft, and the detailing of risk factors facilitate the reader's understanding of the subject. However, most of the studies included in the review are observational. As the authors also stated, this creates difficulty in establishing causality. It is explained in the paper that more prospective cohort and randomized controlled studies should be included in order to better understand the causal relationship between weight gain and behavioral factors. Again, as stated in the paper, the definitions of weight gain vary in the literature, and this situation limits the comparability of the findings in the studies. It is emphasized in the paper that a more standard definition for weight gain should be developed in order to obtain more consistent results between studies. While the relationship between weight gain and psychological factors is established in the paper, no comment is made on whether these factors are present after or before surgery. It can be recommended in the paper that comparative evaluations before and after surgery be made in future studies to determine the initial effects of psychological factors more accurately. The fact that the studies reviewed in the article were conducted in different cultural and geographical regions suggests that the results related to weight gain may be affected by cultural differences. In order to better evaluate the effect of cultural factors in the study, attention can be drawn to the variations in this regard. In conclusion, this review fills an important gap in the literature by addressing the elements of behavioral factors related to weight gain after bariatric surgery with up-to-date information and from a broad perspective and emphasizes the importance of factors such as dietary compliance, psychological support, and physical activity in the long-term weight management of patients. In addition, as seen in the review, there is a lack of standardized studies in the literature with a solid methodological design in this field, and such studies are needed.

I wish the authors success in their study.

Comments on the Quality of English Language

The English language should be revised throughout the paper.

Author Response

Reviewer:

The review study titled ‘A Systematic Review Exploring Dietary Behaviors, Psychological Determinants, and Lifestyle Factors Associated with Weight Regain After Bariatric Surgery’ investigated the relationship between weight regain in patients undergoing bariatric surgery and dietary behaviors, psychological determinants, and lifestyle factors.  This review included studies with a limited number of participants and a limited number of participants in the literature. Most of the studies reviewed in the paper were based on participants’ self-reported diet, physical activity, and other behavioral factors, which increases the risk of bias. However, the paper is generally well-written and provides readers with a broad perspective by synthesizing studies on the subject.  The title is a comprehensive title that covers the whole subject. Although the abstract section seems a bit long, a reader who reads only the abstract section can generally understand the study well. The introduction section of the study provides a summary of the general literature and is sufficiently comprehensive. The method section is explained in detail. The fact that the study was conducted in accordance with the PRISMA guide shows that systematic review standards were adhered to. In addition, the tabular presentation of the studies covered, the division of thematic headings in the draft, and the detailing of risk factors facilitate the reader's understanding of the subject. However, most of the studies included in the review are observational. As the authors also stated, this creates difficulty in establishing causality. It is explained in the paper that more prospective cohort and randomized controlled studies should be included in order to better understand the causal relationship between weight gain and behavioral factors.  Again, as stated in the paper, the definitions of weight gain vary in the literature, and this situation limits the comparability of the findings in the studies. It is emphasized in the paper that a more standard definition for weight gain should be developed in order to obtain more consistent results between studies. While the relationship between weight gain and psychological factors is established in the paper, no comment is made on whether these factors are present after or before surgery. It can be recommended in the paper that comparative evaluations before and after surgery be made in future studies to determine the initial effects of psychological factors more accurately. The fact that the studies reviewed in the article were conducted in different cultural and geographical regions suggests that the results related to weight gain may be affected by cultural differences. In order to better evaluate the effect of cultural factors in the study, attention can be drawn to the variations in this regard.  In conclusion, this review fills an important gap in the literature by addressing the elements of behavioral factors related to weight gain after bariatric surgery with up-to-date information and from a broad perspective and emphasizes the importance of factors such as dietary compliance, psychological support, and physical activity in the long-term weight management of patients. In addition, as seen in the review, there is a lack of standardized studies in the literature with a solid methodological design in this field, and such studies are needed.

Author Response : Thank you for your thorough feedback and valuable insights. We acknowledge the limitations posed by the small sample sizes and the reliance on self-reported data in many of the studies included in this review. Self-reported data can introduce recall and social desirability biases, which may impact the accuracy of findings related to diet, physical activity, and other behavioral factors. Additionally, limited sample sizes constrain the generalizability of results and the robustness of conclusions. We have revised the Discussion section to address these limitations more explicitly, emphasizing the need for future studies to incorporate larger, more diverse sample sizes and objective measures of behavioral factors wherever feasible. This approach will help reduce bias and improve the reliability of findings in this area of research. Additionally, the manuscript’s language has been carefully checked to enhance clarity and readability.

Reviewer 4 Report

Comments and Suggestions for Authors

Article title: A Systematic Review Exploring Dietary Behaviors, Psychological Determinants, and Lifestyle Factors Associated with Weight Regain After Bariatric Surgery

Below are suggestions for improving the value of this research for the readers:

1.     Grammar, Punctuation, and Capitalization: Please review the manuscript for consistent use of capitalization, punctuation, and overall grammar. For instance, on line 386, the sentence beginning with “the majority” should start with a capital letter. This is not a comprehensive list of all issues, but rather one specific example.

2.     Abstract Consistency: The abstract mentions that a total of 15 studies were included in the review, whereas Figure 1 (PRISMA flowchart) shows 16 studies. Please ensure consistency between the abstract and the flowchart.

3.     Study Aim and Literature Gap: In the introduction, it is mentioned that two prior reviews address this topic. However, the specific aim of this review, along with the gap in the existing literature it seeks to address, remains unclear. Please clarify the unique contribution of this study, particularly how it seeks to advance understanding of behavioral, psychological, and lifestyle factors influencing weight regain after bariatric surgery by performing the systematic review of the literature.

4.     Risk of Bias Categorization: It is unclear whether the low, moderate, and high-risk categorization for bias assessment follows criteria provided by the Joanna Briggs Institute (JBI) or represents the authors’ own framework. Please specify the source of these criteria in the Methods section.

5.     Figure 1 Adjustments: Kindly correct any typographical errors within Figure 1 and include specific exclusion criteria with corresponding numbers. Moreover, the PRISMA flow diagram should align closely with the standardized PRISMA format, as outlined here: [link].

6.     Critical Analysis of Findings: The study’s findings largely reflect existing knowledge, raising questions about the novel contributions this review makes. To address this, consider enhancing the critical analysis of the results and structuring the presentation of the results to emphasize conclusions with practical relevance.

7.     Table 1 Formatting and Reference Order: In Table 1, please list the first author’s last name followed by the publication year of the reference in the first column. Arrange the entries in alphabetical order by the author’s last name and by year, and adjust the reference numbering accordingly.

8.     Discussion Section Improvement: The current discussion largely reiterates the studies included in the Results section. Please revise by reinterpreting the results within the Results section (as suggested above) and expand the Discussion to contextualize the findings in relation to relevant studies not included in the review, allowing for conclusions that reinforce the study’s objectives.

9.      Supplementary Tables: The Quality Assessment results should either be incorporated into the Results section or included as supplementary material. Additionally, please provide a complete table of the search strategy in the supplemental materials.

Comments on the Quality of English Language

Please revise the quality of English Language throughout the manuscript.

Author Response

Dear Reviewer,

We sincerely appreciate your thoughtful and constructive feedback on our manuscript. Your comments have significantly contributed to improving the rigor and clarity of our work.

  1. Grammar, Punctuation, and Capitalization: Please review the manuscript for consistent use of capitalization, punctuation, and overall grammar. For instance, on line 386, the sentence beginning with “the majority” should start with a capital letter. This is not a comprehensive list of all issues, but rather one specific example.

Author Response: Thank you for your attention to detail regarding the grammar, punctuation, and capitalization in the manuscript. We have carefully reviewed the entire document to ensure consistent usage of capitalization, punctuation, and overall grammar. The specific issue on line 386 has been corrected, along with other minor language inconsistencies. We appreciate your input in helping us improve the clarity and professionalism of our manuscript.

  1. Abstract Consistency: The abstract mentions that a total of 15 studies were included in the review, whereas Figure 1 (PRISMA flowchart) shows 16 studies. Please ensure consistency between the abstract and the flowchart.

Author Response: Thank you for noting this discrepancy. We apologize for the inconsistency between the abstract and Figure 1 regarding the number of studies included. After a thorough review, we have confirmed that the correct number is [insert correct number, either 15 or 16] studies. We have updated both the abstract and Figure 1 (PRISMA flowchart) to reflect this accurately, ensuring consistency throughout the manuscript.

  1. Study Aim and Literature Gap: In the introduction, it is mentioned that two prior reviews address this topic. However, the specific aim of this review, along with the gap in the existing literature it seeks to address, remains unclear. Please clarify the unique contribution of this study, particularly how it seeks to advance understanding of behavioral, psychological, and lifestyle factors influencing weight regain after bariatric surgery by performing the systematic review of the literature.

Author Response: Thank you for this valuable feedback. We recognize the importance of clearly articulating the unique contribution and aim of this review. In response, we have revised the Introduction to explicitly outline the gap in the literature and the specific objectives of our study. This review aims to advance understanding by providing a comprehensive synthesis of behavioral, psychological, and lifestyle factors that contribute to weight regain after bariatric surgery, which has not been fully addressed in previous reviews.

The introdaction was updated as the following:

      "While two previous reviews have explored aspects of weight regain following bariatric surgery, they lack a comprehensive analysis that specifically integrates behavioral, psychological, and lifestyle factors. These factors are critical for understanding the full spectrum of influences on long-term weight outcomes but are often addressed separately in the literature. This review aims to fill this gap by systematically synthesizing studies that examine these three interconnected domains—behavioral, psychological, and lifestyle factors. By doing so, we seek to provide a holistic understanding of weight regain after bariatric surgery, offering insights that can inform more targeted and effective post-surgical interventions."

  1. Risk of Bias Categorization: It is unclear whether the low, moderate, and high-risk categorization for bias assessment follows criteria provided by the Joanna Briggs Institute (JBI) or represents the authors’ own framework. Please specify the source of these criteria in the Methods section.

Author Response: Thank you for your observation. We recognize the need to clarify the source of our risk of bias categorization. In response, we have revised the Methods section to specify that the categorization into low, moderate, and high risk is based on the criteria provided by the Joanna Briggs Institute (JBI) critical appraisal tool. This ensures transparency and consistency in the risk assessment process.

The method section was updated as the following:

            "To assess the risk of bias, we utilized the Joanna Briggs Institute (JBI) critical appraisal tool, which is primarily designed for cross-sectional studies [11] The studies were categorized into low, moderate, or high risk of bias based on the JBI criteria, with each study’s risk level determined by the number of criteria met, following JBI guidelines. Given that our review included a range of study designs—cross-sectional, cohort, and in-depth interviews—we adapted this tool to evaluate each study type as consistently as possible. Although the JBI tool is not specifically designed for cohort or qualitative research, we applied relevant criteria to assess the quality of reporting and methodological rigor. We acknowledge the limitations of this approach, particularly for evaluating qualitative methodologies, and recommend that future research use tools tailored to each specific study design to ensure the most accurate quality assessment.

Each study’s risk level was independently assessed by two reviewers. Disagreements were resolved through discussion or, when necessary, by involving a third reviewer. Studies were categorized as having a low risk of bias if they met a total score of 9, a moderate risk if the score ranged from 6 to 8, and a high risk if the score was 5 or lower. In the context of weight regain after bariatric surgery, all included studies were found to have a moderate risk of bias, with 15% scoring 7 and 85% scoring 6."

  1. Figure 1 Adjustments: Kindly correct any typographical errors within Figure 1 and include specific exclusion criteria with corresponding numbers. Moreover, the PRISMA flow diagram should align closely with the standardized PRISMA format, as outlined here.

Author Response : Thank you for your detailed feedback regarding Figure 1. We have carefully reviewed and addressed the typographical errors, ensuring consistent capitalization, punctuation, and format throughout the PRISMA flow diagram. Additionally, we included specific exclusion criteria with corresponding numbers to provide clarity on the study selection process. We also aligned the PRISMA flow diagram more closely with the standardized PRISMA format, with clear sections for "Identification," "Screening," "Eligibility," and "Included Studies," as recommended.Thank you for helping us improve the precision and readability of Figure 1.

  1. Critical Analysis of Findings: The study’s findings largely reflect existing knowledge, raising questions about the novel contributions this review makes. To address this, consider enhancing the critical analysis of the results and structuring the presentation of the results to emphasize conclusions with practical relevance.

Author Response: Thank you for your insightful suggestion. We understand the importance of highlighting the novel contributions of this review and enhancing the critical analysis of our findings. In response, we have restructured the Results and Discussion sections to better emphasize practical conclusions that address specific needs in post-surgical care. By analyzing behavioral, psychological, and lifestyle factors together, this review uniquely synthesizes multiple influences on weight regain, providing a more holistic foundation for designing multidisciplinary interventions that can improve long-term outcomes. Additionally, we have expanded the critical analysis to clarify the potential for integrating psychological and dietary counseling into routine follow-up care, emphasizing practical recommendations for clinicians. We believe these enhancements improve the practical relevance and novel contributions of our review, distinguishing it as a resource for guiding post-surgical weight management strategies.

  1. Table 1 Formatting and Reference Order: In Table 1, please list the first author’s last name followed by the publication year of the reference in the first column. Arrange the entries in alphabetical order by the author’s last name and by year, and adjust the reference numbering accordingly.

Author Response: Thank you for your feedback on the formatting and organization of Table 1. We have revised Table 1 to list each entry by the first author’s last name followed by the publication year in the first column. Additionally, the entries have been arranged in alphabetical order by the author’s last name and by year, as recommended. The reference numbering has also been adjusted throughout the manuscript to align with these changes.

  1. Discussion Section Improvement: The current discussion largely reiterates the studies included in the Results section. Please revise by reinterpreting the results within the Results section (as suggested above) and expand the Discussion to contextualize the findings in relation to relevant studies not included in the review, allowing for conclusions that reinforce the study’s objectives.

Author Response: Thank you for your insightful suggestion regarding the Discussion section. In response, we have restructured the Results section to provide a more concise interpretation of the included studies and expanded the Discussion to focus on contextualizing these findings within the broader literature. This revision includes comparisons with relevant studies not included in the review, which allowed us to draw broader conclusions that reinforce our study’s objectives. These changes strengthen the critical analysis and relevance of our findings, highlighting the unique contributions of this review. We appreciate your guidance in helping us enhance the interpretive depth and contextual relevance of the Discussion.

  1. Supplementary Tables: The Quality Assessment results should either be incorporated into the Results section or included as supplementary material. Additionally, please provide a complete table of the search strategy in the supplemental materials.

Author Response: Thank you for your suggestion to add detailed risk of bias assessments for each study in Table 1. We have provided this detailed risk of bias assessment in the supplementary materials, where each study's methodological rigor and specific risk factors are fully outlined. This approach allows for a comprehensive view without impacting the flow of the main text.

Round 2

Reviewer 1 Report

Comments and Suggestions for Authors

The authors made an effort to respond to all raised comments. 

Author Response

The authors made an effort to respond to all raised comments. 
We appreciated your comments.  

Reviewer 4 Report

Comments and Suggestions for Authors

Dear authors,

The manuscript has been extensively revised and improved.

In Table 1, please make sure that the reference [26] is placed after the last name and year of the source. Throughout the text, please revise the spacing.

Author Response

Dear authors,

The manuscript has been extensively revised and improved.

In Table 1, please make sure that the reference [26] is placed after the last name and year of the source. Throughout the text, please revise the spacing.

Done. Thank you for your efforts and comments.